# VEBench: Towards Comprehensive and Automatic Evaluation for Text-guided Video Editing

## Abstract

Video editing task has gained widespread attention in recent years due to their practical applications and rapid advancements, driven by the emergence of diffusion techniques and Multi-modal Large Language Models (MLLMs). However, current automatic evaluation metrics for video editing are mostly unreliable and poorly aligned with human judgments. As a result, researchers heavily rely on human annotation for evaluation, which is not only time-consuming and labor-intensive but also difficult to ensure consistency and objectivity. To address this issue, we introduce **VEBench**, the largest-ever video editing meta-evaluation benchmark to evaluate the reliability of automatic metrics. It includes 152 video clips and 962 text prompts, from which 160 instances are sampled to generate 1,280 edited videos using 8 open-source video editing models, accompanied by human annotations. Especially, the text prompts are first crafted using GPT-4, followed by manual review and careful categorization based on editing types for a systematic evaluation. Our human annotations cover 3 criteria: *Textual Faithfulness*, *Frame Consistency*, and *Video Fidelity*, ensuring the comprehensiveness of evaluation. Since human evaluation is costly, we also propose **VEScore**, employing MLLMs as evaluators to assess edited videos from the criteria above. Experiments show that the best-performing video editing model only reaches an average score of 3.18 (out of a perfect 5), highlighting the challenge of VEBench. Besides, results from more than 10 MLLMs demonstrate the great potential of utilizing VEScore for automatic evaluation. Notably, for Textual Faithfulness, VEScore equipped with LLaVA-OneVision-7B achieves a Pearson Correlation score of 0.48, significantly outperforming previous methods based on CLIP with the highest score of 0.21. The dataset and code will be released upon acceptance.

## 1 Introduction

Video editing has witnessed significant improvements in recent years, with the burgeoning of diffusion techniques (Ho et al., 2020; Rombach et al., 2022) and Multi-modal Large Language Models (MLLMs) (OpenAI, 2024; Reid et al., 2024). Notably, most studies in this field focus on text-guided video editing (Wu et al., 2023a; Yang et al., 2023), which offers a straightforward and intuitive editing process by simply modifying the video caption.

Despite these advancements, the evaluation of video editing still lacks reliable metrics. Current automatic evaluation methods fall short in both comprehensiveness and robustness, due to the inherent complexity of assessing the quality of edits. On the one hand, it's evident that there is a lack of a unified standard for these evaluation methods. The varied criteria used across studies, ranging from textual alignment to temporal consistency, hinder the systematic comparison of different works (Qi et al., 2023; Wu et al., 2023a). On the other hand, existing CLIP-based metrics (Hessel et al., 2021; Parmar et al., 2023) often poorly align with human judgments, leading to discrepancies in evaluation results (Wu et al., 2023a; Qi et al., 2023; Geyer et al., 2023; Wang et al., 2023). Consequently, the evaluation of video editing often necessitates substantial reliance on human annotations, which are time-consuming, labor-intensive, and prone to inconsistency due to subjective interpretations by different annotators.

To tackle these challenges, we propose **VEBench** in this paper. It serves as a comprehensive meta-evaluation benchmark to aid in developing more effective automatic evaluation systems for video editing. As shown in Figure 1, the establishment of VEBench goes through the following three steps:

(1) **Data Collection**. We gather video clips from the DAVIS dataset (Pont-Tuset et al., 2017; Caelles et al., 2018; 2019) and develop an automatic method to efficiently generate a diverse range of text prompts. Specifically, we notice that the primary focus of existing text prompts is often on entities within video captions (Wu et al., 2023a; Yang et al., 2023). To leverage this, we utilize GPT-4V to generate video captions as source prompts, and subsequently create text prompts by employing an LLM to replace the identified entities within these captions. This process is then followed by manual reviews to ensure the quality of prompts. Through the above steps, we collect instances for the subsequent data taxonomy, with each instance containing a video, video caption, and text prompt.

(2) **Data Taxonomy**. We then categorize the above instances based on editing type (*Content* or *Style*) and the number of editing targets (*Single* or *Multiple*), enabling in-depth analyses of model capabilities from multiple perspectives. Content Editing primarily focuses on modifying specific entities within the video, such as animals or objects. Whereas, Style Editing is concerned with transforming the overall artistic style of the video, such as converting the original video into Van Gogh's style or comic style. The number of editing targets is closely related to difficulty levels, with Multiple-Target editing requiring the models to precisely follow the text prompts, meanwhile producing natural videos.

(3) **Data Annotation**. We finally propose three evaluation criteria to comprehensively assess the edited videos: *Textual Faithfulness*, *Frame Consistency*, and *Video Fidelity*. Utilizing these criteria, we engage annotators to evaluate edited videos, generated by a wide range of models including 8 open-source video editing models. This approach not only facilitates a systematic analysis of these models but also enables a meta-evaluation of current and future video editing metrics.

Furthermore, we propose an automatic scoring system, **VEScore**, leveraging the capabilities of various MLLMs. This system is inspired by recent studies (Fu et al., 2023; Yujie et al., 2023) that effectively utilize (M)LLMs to assess tasks challenging for traditional metrics. Given that current MLLMs have acquired extensive world knowledge from vast training corpora and exhibit improved alignment with human preferences through visual instruction tuning, VEScore serves as a more robust and scalable automatic method compared to traditional metrics.

Experiments on various video editing models and automatic evaluation metrics show that, VEBench poses significant challenges, with the top-performing model achieving an average score of 3.18 out of 5 across the three criteria. More analyses indicate that existing models struggle with editing types involving fine details, such as Human Editing and Animal Editing. Besides, they perform significantly worse on Multiple-Target editing, with an average drop of 2 points from Single-Target to Multiple-Target categories. Finally, we demonstrate the great potential of employing VEScore for automatic evaluation after investigating more than 10 MLLMs. Though there is still much room for improvement, we observe notable improvements in alignments with human preference over traditional metrics across all criteria.

In summary, we contribute VEBench and VEScore to foster the development of comprehensive and automatic evaluation for video editing. We suggest future research to explore their methods on VEBench and further enhance VEScore by employing more robust MLLMs and refining the instructions through prompt engineering.

## 2 RELATED WORK

### 2.1 TEXT-GUIDED VIDEO EDITING

**Benchmarks**   In the development process of text-guided video editing, there has been a notable lack of a unified framework for evaluation. Previous studies typically select several video clips from DAVIS (Wu et al., 2023a) or YouTube (Molad et al., 2023) and manually design text prompts for further inference and evaluation. TGVE (Wu et al., 2023b) summarizes previous evaluation methods and introduces a new benchmark. However, it is limited by the data scale (comprising 76 videos and

304 text prompts) and coverage of editing types (involving only 4 types). In comparison, VEBench offers richer and more diverse instances for evaluation. Besides, it serves as a meta-evaluation benchmark for developing automatic scoring systems.

**Evaluation Metrics**   There are three key automatic evaluation metrics commonly employed in recent years for text-guided video editing task:

- **Textual Alignment** (Wu et al., 2023a). This metric assesses the degree of alignment between the text prompts and the edited video frames by computing the average similarity.

- **Frame Accuracy** (Parmar et al., 2023). This metric measures the percentage of frames in edited videos that have a higher CLIP similarity to the text prompt than to the source prompt, indicating the effectiveness of per-frame edits.

- **Temporal Consistency** (Qi et al., 2023). It evaluates the smoothness of edits across consecutive frames by computing the similarity between consecutive frames.

However, due to the poor reliability of these automatic metrics, previous studies have to rely on **User Preference** (Wu et al., 2023a; Geyer et al., 2023; Wang et al., 2023; Ceylan et al., 2023; Li et al., 2024b; Kara et al., 2024) for more accurate evaluation. Nonetheless, they often suffer from higher costs and inconsistency across different studies due to the subjectivity of annotators.

**Methods**   Most approaches for text-guided video editing are built upon diffusion models, which demonstrate remarkable success in image synthesis (Ho et al., 2020; Rombach et al., 2022; Ramesh et al., 2022) and image editing (Zhang et al., 2023b; Tumanyan et al., 2023). Dreamix (Molad et al., 2023) is the first to utilize video diffusion models on this task through fine-tuning. However, considering computational constraints and overall video quality, later studies (Wu et al., 2023a; Qi et al., 2023) have shifted towards leveraging image diffusion models for video editing. Besides, recent research (Ceylan et al., 2023; Li et al., 2024b; Kara et al., 2024) further focuses on maintaining the temporal consistency between adjacent frames, effectively ensuring the smoothness and coherence of produced videos.

## 2.2 META EVALUATION

Meta evaluation aims to assess the reliability of automatic metrics by examining how closely they align with human judgments (Fabbri et al., 2021; Fu et al., 2023). There are three primary correlation measures used in meta-evaluation: **Pearson Correlation** ($r$) (Freedman et al., 2007), which determines the linear association between two variables; **Spearman's Rho** ($\rho$) (Zar, 2005), which evaluates the monotonic connection between two variables; and **Kendall's Tau** ($\tau$) (Kendall, 1938), which assesses the ordinal relationship between two variables. Previous studies have applied meta evaluation to tasks such as text summarization (Wang et al., 2020; Fabbri et al., 2021; Gopalakrishnan et al., 2023), machine translation (Freitag et al., 2021), and image-text matching task (Yujie et al., 2023), contributing to the development of more comprehensive evaluation metrics that better correlate with human judgments.

## 2.3 (M)LLMs AS EVALUATORS

Recently, (M)LLMs have greatly advanced the development of various research fields (OpenAI, 2023; Team et al., 2023). One popular direction is utilizing (M)LLMs as automatic scoring systems (Fu et al., 2023; Liu et al., 2023; Chan et al., 2023; Zheng et al., 2023; Yujie et al., 2023). By prompting them to follow evaluation criteria, (M)LLMs can provide judgments that align more closely with human preferences than traditional metrics. Thus, this approach proves particularly valuable for complex tasks that heavily rely on human evaluations, such as open-ended generation (Zheng et al., 2023) and text-to-image generation (Yujie et al., 2023). For text-guided video editing, using MLLMs for automatic evaluation remains unexplored, due to the absence of a unified evaluation benchmark and the inherently challenging nature of the task. To the best of our knowledge, our work is the first attempt to explore this approach for text-guided video editing evaluation and also provide the best configuration after investigating more than 10 MLLMs.

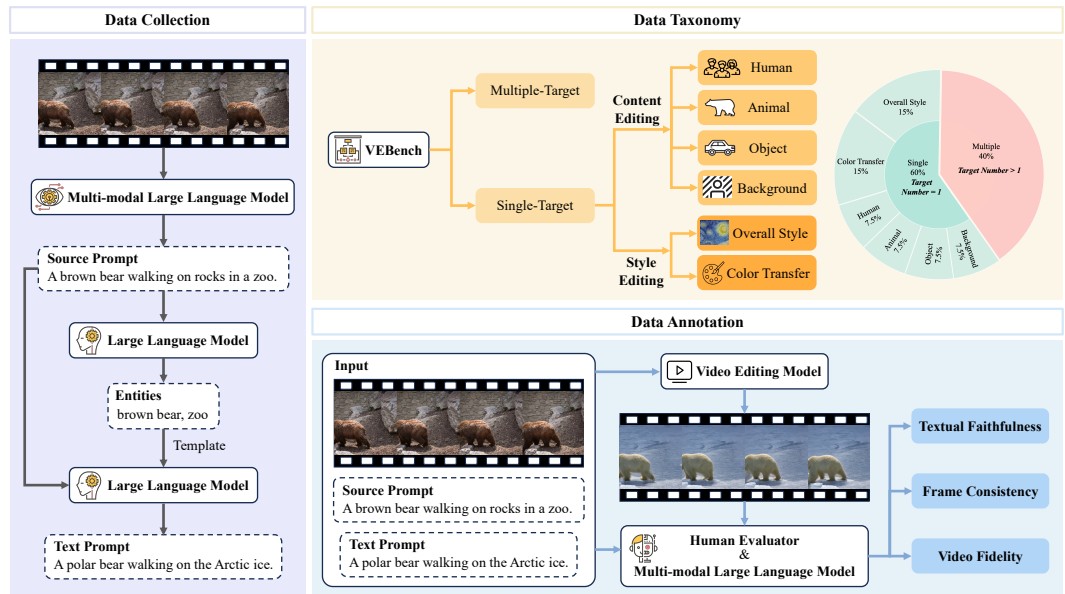

Figure 1: The construction process of VEBench involves three main steps: Data Collection, where high-quality videos and text prompts are gathered; Data Taxonomy, where instances are categorized by editing types and target numbers; and Data Annotation, where human evaluations are conducted based on three criteria: Textual Faithfulness, Frame Consistency, and Video Fidelity.

## 3    VEBENCH

In this section, we describe the three steps of constructing VEBench: data collection (Section 3.1), data taxonomy (Section 3.2), and data annotation (Section 3.3).

### 3.1    DATA COLLECTION

The data collection process of VEBench consists of *Video Collection* and *Text Prompt Generation* substeps. All data produced in each substep is manually reviewed to ensure the quality, detailed in Appendix A.

**Video Collection**    Following previous research (Wu et al., 2023a), we gather video clips from the publicly available DAVIS datasets (Pont-Tuset et al., 2017; Caelles et al., 2018; 2019), which are specifically tailored for video object segmentation tasks. By collating data from DAVIS 2017 to DAVIS 2019 and removing duplicates, we obtain 210 video clips in total. Subsequently, we conduct a careful manual review to filter out videos featuring indistinct objects or of substandard quality and finally obtain 152 high-quality videos suitable for editing. Given the current video editing models do not support excessively long videos or high-resolution inputs, we resize all videos to a resolution of $480 \times 480$ and uniformly sample 25 frames from the original videos.

**Text Prompt Generation**    As illustrated in the left of Figure 1, we develop an entity-centric automatic generation approach to produce diverse text prompts. This draws inspiration from prior research on object, background, and animal editing (Wu et al., 2023a; Huang et al., 2024), where most edited targets typically manifest as entities within the corresponding video captions. To be specific, we first adopt GPT-4V to obtain captions for each video, which also serve as source prompts in text-guided video editing task. To ensure quality, a manual review is conducted to eliminate any inaccuracies in the generated captions. Next, we utilize GPT-4 to extract entities from these captions. Then, text prompts are crafted by instructing GPT-4 based on the extracted entities, as shown in Appendix B.1. Especially, the given instruction encourages GPT-4 to maintain the original semantic meaning of captions, thereby enhancing the suitability for video editing task.

## 3.2 DATA TAXONOMY

One of the challenges in video editing stems from the diversity of editing types and their combinations, which significantly escalate the difficulty. As current video editing models underperform on complex video editing tasks, analyzing their performance across various editing types can provide valuable insights for improvement. To this end, we perform data taxonomy on the collected instances based on their editing types and number of targets as shown in the upper-right of Figure 1.

Table 1: Examples of different editing types with original video captions and corresponding text prompts. The first six rows represent Single-Target Editing categories, including Human Editing, Animal Editing, Object Editing, Background Editing, Overall Style Editing, and Color Transfer. The final row illustrates an example of Multiple-Target Editing. The highlighted text indicates the editing targets.

| Category | Video Caption | Text Prompt |
|---|---|---|
| Human Editing | A *man* in shorts is fixing a bike in the room | A *robot* in shorts is fixing a bike in the room |
| Animal Editing | A *dog* is running on a grassy path in a yard with a fence | A *cat* is running on a grassy path in a yard with a fence |
| Object Editing | A *suv* driving down a winding road with mountains in the background | A *jeep* driving down a winding road with mountains in the background |
| Background Editing | A suv driving down a winding road with *mountains* in the background | A suv driving down a winding road with a *volcano* in the background |
| Overall Style Editing | A man in shorts is fixing a bike in the garage | A man in shorts is fixing a bike in the garage, *in oil painting style* |
| Color Transfer | A man in shorts is fixing a bike in the garage | A man in shorts is fixing a bike in the room, *in grayscale* |
| Multiple-Target Editing | A *suv* driving down a winding road with *mountains* in the background | A *jeep* driving down a winding road with a *volcano* in the background, in *Van Gogh style* |

**Editing Type**   Inspired by earlier studies (Wu et al., 2023a; Huang et al., 2024), we categorize the instances into *Content Editing* and *Style Editing*. Content Editing concentrates on local modifications to specific objects within videos, encompassing *Human Editing*, *Animal Editing*, *Object Editing*, and *Background Editing*. In contrast, Style Editing emphasizes global changes to all content involved in videos, including *Overall Style Editing* and *Color Transfer*. Examples of these editing types are shown in Table 1.

**Target Number**   Most previous studies (Wu et al., 2023b; Qi et al., 2023; Kara et al., 2024) mainly focus on Single-Target editing, ignoring the practical requirement of editing multiple targets in a video. To tackle this limitation, we further sort the data into *Single* and *Multiple* categories. For instance, the case illustrated in Figure 1 belongs to the Multiple-Target category, because it involves two specific editing targets, "*brown bear*" and "*zoo*", which are asked to be edited as "*polar bear*" and "*Arctic ice*", respectively. Intuitively, editing multiple targets is notably more challenging for current models, which may require better mechanisms and training strategies to reach promising performance. By providing these evaluation subsets, we can foster the advancement of video editing models in tackling more intricate editing tasks, finally moving towards practical video editing applications.

Through the above substeps, we ultimately collect 962 text prompts, which are further categorized based on *Content Editing* and *Style Editing*, as well as *Single-Target* and *Multiple-Target* edits. The detailed data statistics can be found in Appendix B.2.

## 3.3 DATA ANNOTATION

The further goal of VEBench is to facilitate the meta-evaluation for video editing scoring systems. To this end, we sample 160 instances, which are then inferred by 8 different video editing models,

resulting in a total of 1,280 edited videos. Then, we hire 4 professional annotators to provide human evaluation for these generated videos, as illustrated in the lower-right of Figure 1. Each annotator is assigned 30 instances per day to avoid fatigue and we also conduct regular spot checks to ensure annotation quality. We consider the following criteria to comprehensively evaluate each video:

- **Textual Faithfulness**. It evaluates how well the edited video aligns with the text prompt. A higher score indicates that the edited video accurately reflects all the details specified in the text prompt.

- **Frame Consistency**. This metric assesses the continuity between frames in the edited video. A higher score indicates seamless transitions with no noticeable jumps between frames.

- **Video Fidelity**. This measures the quality and visual consistency of the edited video, considering factors such as color, resolution, dynamic range, and motion coherence. A higher score indicates excellent video quality, with no detectable issues in color, sharpness, or continuity, whereas a lower score suggests problems such as blurriness, color distortion, and poor continuity.

The annotated scores for all criteria range from 1 to 5, where 5 denotes a state of flawlessness. Detailed annotation guidelines regarding these criteria are provided in Appendix C.1. As implemented in FETV (Liu et al., 2024b), we select Kendall's Tau ($\tau$), Spearman's Rho ($\rho$), and Krippendorff's Alpha ($\alpha$) (Krippendorff, 2018) for the evaluation of inter-annotator agreements. From Table 2 we observe that the inter-annotator agreements across all metrics and three criteria exceed 0.6, indicating minimal variability among annotators and reinforcing the reliability of the human annotation process.

The detailed human annotation interface, including the display of source prompts, text prompts, and the original and edited videos, can be found in the Appendix C.2.

Table 2: Inter-annotator agreement scores for our proposed three evaluation criteria: Textual Faithfulness, Frame Consistency, and Video Fidelity. Kendall's $\tau$ and Spearman's $\rho$ are averaged across all pairwise correlations between annotators. The ± symbol represents standard deviations.

|  | **Textual Faithfulness** | **Frame Consistency** | **Video Fidelity** |
|---|---|---|---|
| **Kendall's** $\tau$ | $0.64 \pm 0.07$ | $0.65 \pm 0.02$ | $0.61 \pm 0.03$ |
| **Spearman's** $\rho$ | $0.71 \pm 0.08$ | $0.73 \pm 0.02$ | $0.69 \pm 0.03$ |
| **Krippendorff's** $\alpha$ | 0.696 | 0.669 | 0.663 |

## 4 MLLMs AS EVALUATORS FOR TEXT-GUIDED VIDEO EDITING

Due to the intricate nature of this task, previous automatic evaluation metrics are mostly built upon CLIP models (Radford et al., 2021). However, CLIP models are recognized for their limited discernment capabilities, particularly in complex scenarios (Tong et al., 2024). Besides, these intuitive approaches often lack comprehensiveness and have not undergone strict verification, leading to the poor reliability of these metrics.

Inspired by previous research on the effectiveness of using (M)LLMs to evaluate various tasks (e.g., open-ended questions) (Fu et al., 2023; Chan et al., 2023; Yujie et al., 2023), we introduce MLLMs as evaluators to address this challenge.

As illustrated in Figure 2, for each instance, an MLLM processes the instruction, source prompt, text prompt, edited video, and the detailed guideline of evaluation criteria to generate the corresponding scores. Notably, not all MLLMs support video inputs. For the models that do, we directly input the videos. For the remaining models, we uniformly sample frames from the videos and use them as multiple image inputs. Then, using three separate queries, we can derive three scores for each criterion based on the responses of MLLM. By leveraging its capabilities in visual perception and instruction-following, we derive detailed scores for each criterion, akin to those provided by human annotators.

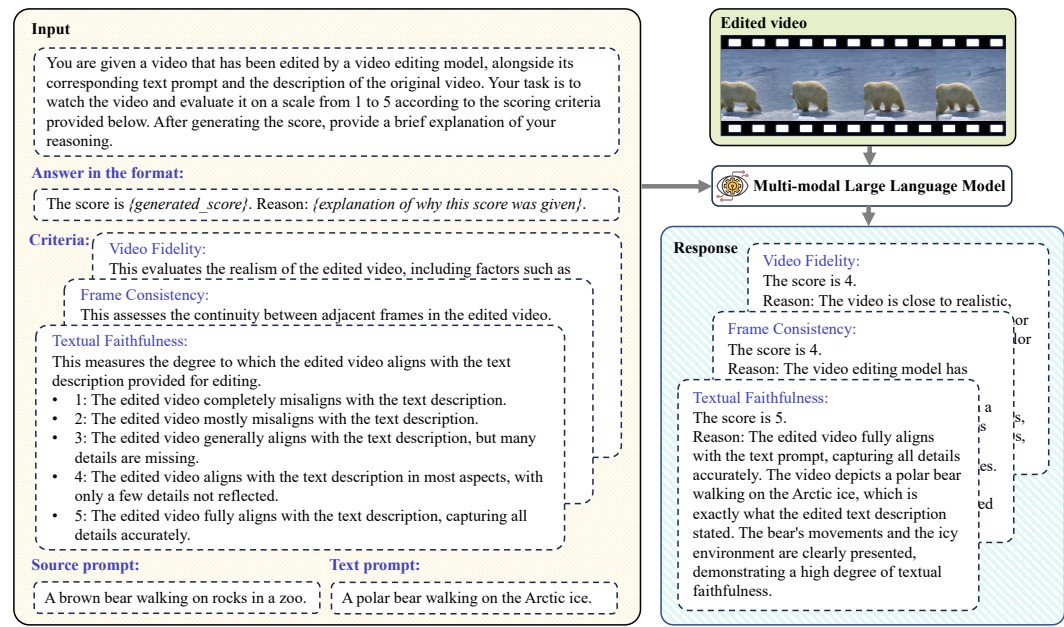

Figure 2: Automatic scoring process utilizing MLLMs based on three evaluation criteria: Textual Faithfulness, Frame Consistency, and Video Fidelity. The MLLM evaluates the instruction, source prompt, text prompt, and edited video according to detailed guidelines, generating corresponding scores.

Table 3: Overall performance of Video Editing Models on VEBench and its Single / Multiple-Target subsets. TF, FC, and VF denote Textual Faithfulness, Frame Consistency, and Video Fidelity, respectively. Avg is the average of these 3 scores.

| Model | VEBench-ALL | | | | VEBench-Single | | | | VEBench-Multiple | | | |
|---|---|---|---|---|---|---|---|---|---|---|---|---|
| | TF | FC | VF | Avg | TF | FC | VF | Avg | TF | FC | VF | Avg |
| VidToMe | 3.35 | 3.10 | 2.93 | 3.13 | 3.99 | 3.61 | 3.41 | 3.67 | 2.39 | 2.34 | 2.22 | 2.32 |
| TokenFlow | 3.31 | **3.15** | 3.08 | **3.18** | **4.02** | 3.67 | 3.60 | 3.76 | 2.24 | 2.37 | 2.29 | 2.30 |
| Text2Video-Zero | 2.73 | 1.67 | 1.59 | 2.00 | 2.97 | 1.89 | 1.76 | 2.20 | 2.37 | 1.34 | 1.34 | 1.68 |
| FateZero | 3.08 | 3.14 | **3.14** | 3.12 | 3.99 | **3.95** | **3.97** | **3.97** | 1.70 | 1.93 | 1.91 | 1.85 |
| Tune-A-Video | 3.21 | 2.54 | 2.29 | 2.68 | 3.69 | 2.92 | 2.60 | 3.07 | 2.49 | 1.98 | 1.82 | 2.09 |
| RAVE | **3.38** | 3.11 | 2.96 | 3.15 | 3.94 | 3.49 | 3.32 | 3.58 | 2.54 | **2.54** | **2.43** | **2.51** |
| vid2vid-zero | 3.07 | 2.10 | 2.09 | 2.42 | 3.51 | 2.32 | 2.33 | 2.72 | 2.41 | 1.77 | 1.73 | 1.97 |
| Pix2Video | 3.39 | 2.75 | 2.62 | 2.92 | 3.81 | 3.03 | 2.88 | 3.24 | **2.75** | 2.33 | 2.23 | 2.44 |

## 5 EXPERIMENTS

### 5.1 MODEL EVALUATION

As mentioned in Section 3.3, our human annotations cover 8 mainstream open-source video editing models, including VidToMe (Li et al., 2024b), TokenFlow (Geyer et al., 2023), Text2Video-Zero (Khachatryan et al., 2023), FateZero (Qi et al., 2023), Tune-A-Video (Wu et al., 2023a), RAVE (Kara et al., 2024), vid2vid-zero (Wang et al., 2023), and Pix2Video (Ceylan et al., 2023). We first investigate the performance of these models (Table 3) and then further conduct detailed analyses on different editing categories (Figure 4) of VEBench. Observations are as follows:

***TokenFlow Performs Best Among the Video Editing Models, yet Still Has Much Room to Improve***
As illustrated in Table 3, TokenFlow achieves the highest overall average score of 3.18, succeeded by

RAVE and VidToMe, with respective scores of 3.15 and 3.13. Notably, all these models emphasize inter-frame feature correspondences, which aids in achieving higher frame consistency and video fidelity compared to other models. Nonetheless, they still fall short of perfection, considering the maximum score of 5. This highlights the challenges presented by our VEBench and underscores the necessity for developing more robust video editing models.

***Creating Natural and High-fidelity Videos Poses a Significant Challenge for Video Editing Models*** A successful edit must not only accurately modify the target as specified in the text prompt but also ensure the smoothness and fidelity of the edited video. We observe that nearly all models have considerably lower Frame Consistency and Video Fidelity scores compared to Textual Faithfulness ones. Besides, the same conclusion can be drawn from Figure 3, which shows that for Textual Faithfulness, the most common human annotation score is 4. In contrast, regarding Frame Consistency and Video Fidelity, the score distribution peaks at 2. Some studies (Ceylan et al., 2023; Li et al., 2024b; Kara et al., 2024) have effectively addressed this issue by integrating inter-frame feature correspondences. Popular cutting-edge video generation techniques may also help to improve these parts.

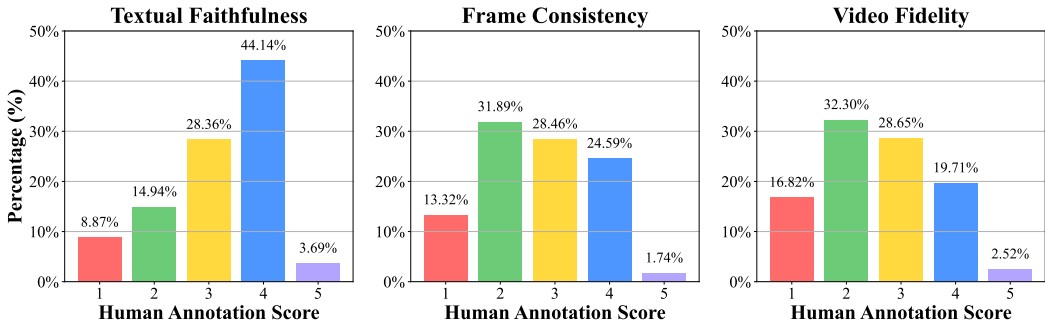

Figure 3: Distribution of human annotation scores on VEBench across three criteria: Textual Faithfulness, Frame Consistency, and Video Fidelity. Different colors are used to represent scores from 1 to 5, with the same color scheme applied consistently across all three charts for each score.

***Conducting Multiple-Target Editing Presents a Substantial Challenge for Existing Video Editing Models*** As depicted on the right side of Table 3, we further investigate the performance of these models on VEBench subsets involving either single or multiple editing targets. We observe a significant decline in model performance when transitioning from VEBench-Single to VEBench-Multiple across all models. Particularly, FateZero achieves the best performance on VEBench-Single but experiences a 2-point decrease on VEBench-Multiple. This observation highlights that current models still struggle to effectively follow complex text prompts. Further efforts are needed to address this problem.

***Current Video Editing Models Struggle with Precise Editing*** As illustrated in Figure 4, these models show relatively stronger performance in general style editing tasks, such as Color Transfer, but perform worse when it comes to fine object editing, particularly for Human and Animal. This is primarily due to the fact that these categories require more detailed and substantial modifications, exemplified by editing like "camel to elephant" or "man to woman". This demonstrates that current models still face difficulties in achieving precise and careful editing.

## 5.2 METRIC EVALUATION

Table 4 shows the meta-evaluation results on traditional metrics and our VEScore using various MLLMs on the 1,280 human-annotated instances. The traditional metrics we evaluate include commonly used CLIP-Textual-Alignment (Wu et al., 2023a), CLIP-Frame-Acc (Qi et al., 2023), CLIP-Temporal-Consistency (Wu et al., 2023a), and LAION-Aesthetic-Predictor (Schuhmann et al., 2022), each of which focuses on only one of three proposed criteria. Although prior work did not employ an automatic scoring metric for video fidelity, we use the LAION-Aesthetic-Predictor as a tra-

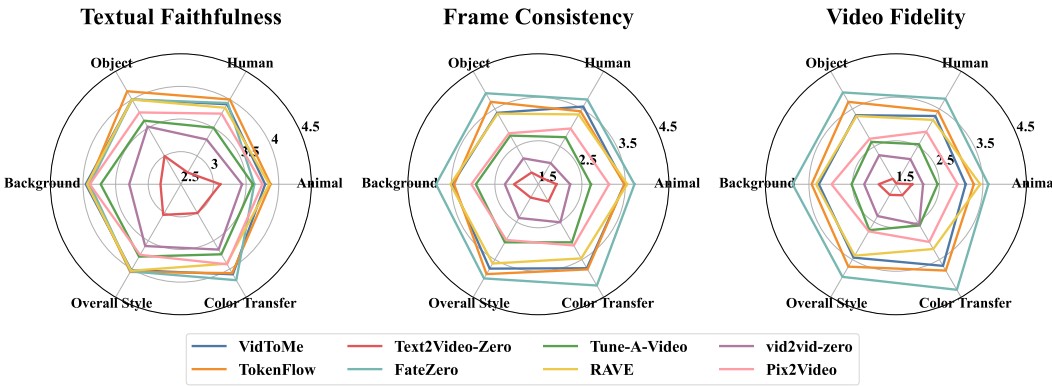

Figure 4: Comparison of different models across six subcategories, evaluated by our proposed three criteria: Textual Faithfulness, Frame Consistency, and Video Fidelity.

Table 4: Overall performance of traditional metrics and MLLMs on our benchmark. Empty symbol (-) indicates that the model is unable to correctly follow the evaluation criteria or not support the evaluation of certain criteria. The metrics $r(\uparrow)$, $\rho(\uparrow)$, and $\tau(\uparrow)$ correspond to Pearson Correlation, Spearman Correlation, and Kendall's Tau, respectively.

| Evaluator | Parameters | Textual Faithfulness | | | Frame Consistency | | | Video Fidelity | | |
|---|---|---|---|---|---|---|---|---|---|---|
| | | $r(\uparrow)$ | $\rho(\uparrow)$ | $\tau(\uparrow)$ | $r(\uparrow)$ | $\rho(\uparrow)$ | $\tau(\uparrow)$ | $r(\uparrow)$ | $\rho(\uparrow)$ | $\tau(\uparrow)$ |
| *Traditional score system* | | | | | | | | | | |
| CLIP-Textual-Alignment | 428M | 0.21 | 0.15 | 0.11 | - | - | - | - | - | - |
| CLIP-Frame-Acc | 428M | 0.07 | -0.01 | 0.00 | - | - | - | - | - | - |
| CLIP-Temporal-Consistency | 428M | - | - | - | 0.26 | 0.27 | 0.19 | - | - | - |
| LAION-Aesthetic-Predictor | 428M | - | - | - | - | - | - | 0.05 | 0.04 | 0.03 |
| *Open-source MLLMs* | | | | | | | | | | |
| TimeChat | 7B | 0.19 | 0.15 | 0.12 | 0.12 | 0.11 | 0.08 | 0.14 | 0.14 | 0.11 |
| Video-LLaMA | 7B | - | - | - | - | - | - | - | - | - |
| Video-LLaMA-2 | 7B | 0.42 | 0.45 | 0.36 | 0.22 | 0.21 | 0.16 | 0.11 | 0.11 | 0.09 |
| Kangaroo | 7B | - | - | - | - | - | - | - | - | - |
| Qwen-VL-Chat | 7B | 0.33 | 0.32 | 0.26 | 0.15 | 0.17 | 0.14 | 0.10 | 0.14 | 0.12 |
| LLaVA-NeXT-Video | 7B | 0.17 | 0.19 | 0.15 | -0.01 | -0.02 | -0.02 | 0.00 | -0.02 | -0.02 |
| LLaVA-NeXT-Video | 32B | 0.47 | **0.48** | **0.39** | **0.27** | **0.28** | **0.23** | **0.27** | **0.28** | **0.22** |
| LLaVA-OneVision | 7B | **0.49** | **0.48** | **0.39** | 0.17 | 0.18 | 0.14 | 0.07 | 0.07 | 0.06 |
| LLaVA-OneVision | 70B | 0.46 | 0.44 | 0.35 | 0.19 | 0.20 | 0.16 | 0.22 | 0.23 | 0.19 |
| VILA | 34B | 0.39 | 0.39 | 0.31 | 0.22 | 0.23 | 0.18 | 0.16 | 0.16 | 0.13 |
| *Closed-source MLLMs* | | | | | | | | | | |
| GPT-4o-0806 | - | 0.36 | 0.33 | 0.26 | **0.35** | **0.35** | **0.27** | 0.24 | 0.27 | 0.22 |
| GPT-4o | - | 0.31 | 0.28 | 0.23 | 0.26 | 0.28 | 0.22 | 0.18 | 0.19 | 0.15 |
| Gemini-pro | - | **0.37** | **0.34** | **0.27** | 0.25 | 0.27 | 0.21 | **0.26** | **0.29** | **0.23** |

ditional metric. This predictor incorporates an MLP based on CLIP and generates a score reflecting the overall aesthetic expressiveness of an image. Following the calculation method of CLIP-Textual-Alignment, we take the average score of all edited video frames as the final score. To be specific, we employ clip-vit-large-patch14 (Radford et al., 2021) for computing traditional CLIP-based metrics. Regarding our method, we investigate a variety of open-source and closed-source MLLMs, including TimeChat (Ren et al., 2024), Video-LLaMA (Zhang et al., 2023a), Video-LLaMA-2 (Cheng et al., 2024), Kangaroo (Liu et al., 2024a), Qwen-VL-Chat (Bai et al., 2023), LLaVA-NeXT-Video (Zhang et al., 2024), LLaVA-OneVision (Li et al., 2024a), VILA (Lin et al., 2023), GPT-4o (OpenAI, 2024), and Gemini-pro (Reid et al., 2024).

***Traditional Metrics Struggle to Align with Human Judgments*** Traditional metrics in video editing tasks demonstrate a marked weakness in correlating with human judgments. The CLIP-Frame-Acc and LAION-Aesthetic-Predictor yield scores around 0 for their respective criteria, which im-

plies a near absence of correlation with human annotations. Although CLIP-Textual-Faithfulness and CLIP-Frame-Consistency show some positive correlations, their correlation scores remain below 0.3, indicating their limited reliability. These results systematically echo the previous findings, underscoring the urgent need for building robust scoring systems.

***VEScore Shows Great Potential for Reliable Evaluation*** VEScore consistently gives better correlation scores compared to traditional metrics when adopting some recent MLLMs, which have undergone better alignment tuning. Specifically, VEScore especially helps in evaluating Textual Faithfulness, even when employing 7B open-source LLaVA-OneVision, achieving an impressive $r$ of 0.49. Besides, LLaVA-OneVision-7B even outperforms closed-source MLLMs (0.49 vs. 0.37 of $r$). This suggests that factors such as instruction tuning data and task-specific optimizations may have a greater influence on performance than the model scale. For Frame Consistency and Video Fidelity, its performance may not be as robust, with scores ranging from $0.2 \sim 0.4$ when using large-scale models (e.g., LLaVA-NeXT-Video-32B and closed-sourced MLLMs). Nevertheless, it still significantly outperforms previous metrics. We believe it has great potential for further improvements by searching for more effective instructions and developing stronger MLLMs.

***Statistics of Failure Cases for VEScore Using Different MLLMs*** As shown in Appendix B.3, both Video-LLaMA and Kangaroo fail to generate responses in the required format, leading to null outputs. Specifically, Video-LLaMA only produces video descriptions, while Kangaroo is restricted to selecting predefined options like A, B, or C. This highlights the importance of the generalization capabilities of MLLMs. Most open-source MLLMs can follow the required format and generate appropriate outputs. In contrast, closed-source MLLMs tend to refuse to answer more frequently. Despite receiving video inputs or extracted frames, closed-source MLLMs frequently struggle, often responding with statements like, *"I'm sorry, I cannot process any information from the real world,"* or *"I'm unable to watch videos, but I can assist with text-based tasks.".* This underscores the challenge of balancing the multi-modal instruction-following abilities, the safety of MLLMs, and the foundational capabilities of LLMs.

## 6    CONCLUSION

In this paper, we introduce **VEBench**, a comprehensive meta-evaluation benchmark designed to assess the reliability of automatic evaluation metrics in text-guided video editing. VEBench comprises 152 high-quality video clips, 962 meticulously crafted text prompts, and human annotations of 1,280 edited videos generated by 8 state-of-the-art open-source video editing models. By categorizing the data based on editing types and the number of targets, we provide a structured framework for in-depth analysis of model capabilities.

Our experiments reveal that existing video editing models still face significant challenges, particularly in ensuring frame consistency and video fidelity, as well as handling complex editing tasks involving multiple targets. Notably, we demonstrate that current automatic evaluation metrics, primarily based on CLIP models, exhibit poor alignment with human judgments.

To address this gap, we propose **VEScore**, leveraging MLLMs as automatic evaluators. Our results from more than 10 prominent MLLMs show that VEScore aligns with human evaluations significantly better than traditional metrics across all criteria: Textual Faithfulness, Frame Consistency, and Video Fidelity. This finding underscores the potential of MLLMs to automate the evaluation process effectively, reducing the reliance on time-consuming and inconsistent human annotations.

We believe that VEBench will serve as a valuable resource for the research community, facilitating the development of more robust video editing models and more reliable automatic evaluation metrics. Future work could explore enhancing the evaluation of MLLMs capabilities further, as well as extending VEBench to include more diverse video content and editing tasks. Additionally, investigating methods to improve frame consistency and video fidelity in video editing models remains a promising direction for advancing the field.

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

## A    MORE DETAILS OF HUMAN QUALITY REVIEW

Throughout the data construction process, we implement human quality reviews at each stage of Data Collection to ensure the reliability and accuracy of the data.

First, we apply a quality filter to the videos obtained from the Video Collection stage. A checklist is used for manual inspection, and any videos that fail to meet the criteria are discarded:

**Video Quality Guidelines:**

1. Does the video feature a clear and identifiable main subject?
2. Is the overall video quality acceptable (i.e., not too poor)?
3. Are camera movements in the video smooth, without being overly abrupt or disorienting?

Next, we assess the captions generated by GPT-4V for quality. As in the previous step, we use a checklist to verify and, if necessary, correct the captions based on the following standards:

**Generated Video Caption Guidelines:**

1. Does the caption accurately describe the core elements of the video?
2. Does the caption provide a comprehensive description of the video's elements, including background and animals?
3. Is the caption grammatically correct?
4. Is the caption overall fluent and coherent?

After refining the captions manually to ensure their quality, we proceed to generate text prompts based on the extracted entities. A final checklist is used to verify the quality of the generated text prompts:

**Generated Text Prompt Guidelines:**

1. Does the generated text prompt accurately modify or replace the target entity's vocabulary?
2. Is the replacement vocabulary suitable for video editing tasks?
3. Is the generated text prompt grammatically correct?
4. Is the text prompt fluent and coherent overall?

## B    MORE DETAILS OF VEBENCH

### B.1    TEMPLATE FOR TEXT PROMPTS

As illustrated in Figure 5, our template for generating text prompts is designed to streamline the process by providing a structured framework. The template consists of three main components: the instructions, the caption, and the entities extracted from the caption.

### B.2    DATA STATISTICS

We provide a detailed summary of the VEBench statistics. As shown in Table 5, VEBench includes 152 video clips with consistent resolution and length, along with 962 corresponding text prompts. These prompts span a variety of editing tasks, ranging from Single-Target editing to more complex Multiple-Target editing, ensuring a diverse set of scenarios for video editing evaluation. For further inference and meta-evaluation, a subset of 160 text prompts is sampled to create a focused test set.

To further illustrate the diversity of VEBench, we also provide word clouds for the Content Editing and Style Editing categories. As illustrated in Figure 6, VEBench encompasses two main categories: Content Editing and Style Editing. There is a noticeable distinction in the data distribution between these two categories. As shown in Figure 6a, content editing primarily involves elements such as *robots*, *cats*, and *beaches*, while Style Editing, as shown in Figure 6b, includes elements like *watercolor*, *Van Gogh*, and *painting*. This demonstrates the intuitive understanding of the prompt distribution and highlights the diversity of our VEBench.

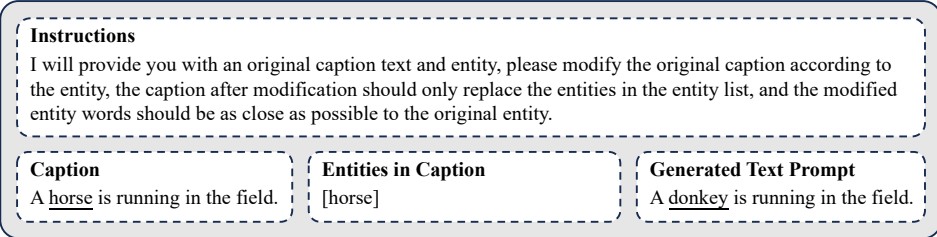

Figure 5: The template used for generating text prompts, consisting of instructions, the original caption, and extracted entities.

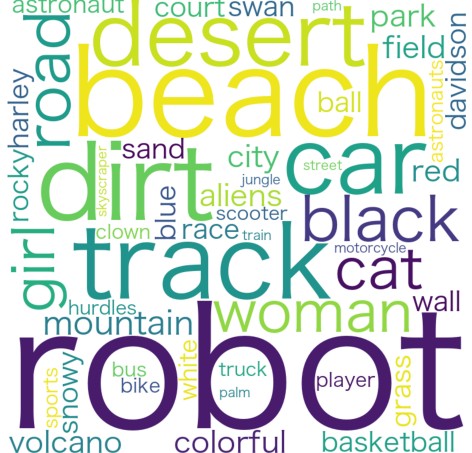

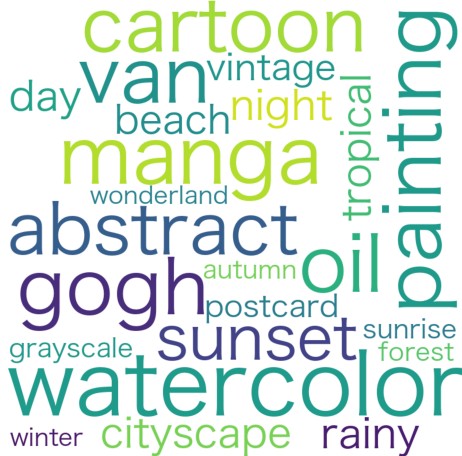

(a) Word cloud of Content Editing category, including Human Editing, Animal Editing, Object Editing, and Background Editing.

(b) Word cloud of Style Editing category, including Overall Style Editing and Color Transfer.

Figure 6: Visualization of key terms from the Content Editing and Style Editing categories in VEBench, highlighting the distribution and emphasis of various editing types.

Table 5: Data statistics of our VEBench.

| Statistic | Number |
|---|---|
| *Video Statistics* | |
| Total video clips | 152 |
| Video resolution | $480 \times 480$ |
| Video Length | 25 frames |
| *Text Prompts Statistics* | |
| Total text prompts | 962 |
| - Single-Target Editing | 658 |
|   - Animal Editing | 43 |
|   - Human Editing | 91 |
|   - Object Editing | 82 |
|   - Background Editing | 138 |
|   - Overall Style Editing | 152 |
|   - Color Transfer | 152 |
| - Multiple-Target Editing | 304 |
| *Text Prompts Statistics for Meta Evaluation* | |
| Total text prompts | 160 |
| - Single-Target Editing | 96 |
|   - Animal Editing | 12 |
|   - Human Editing | 12 |
|   - Object Editing | 12 |
|   - Background Editing | 12 |
|   - Overall Style Editing | 24 |
|   - Color Transfer | 24 |
| - Multiple-Target Editing | 64 |
| Maximum editing target | 5 |
| Minimum editing target | 1 |
| Average editing target | 1.8 |
| Maximum caption length | 29 |
| Minimum caption length | 6 |
| Average caption length | 14.4 |
| Maximum text prompt length | 29 |
| Minimum text prompt length | 7 |
| Average text prompt length | 16.5 |

### B.3 ANALYSIS OF UNMATCHED RESPONSES IN MLLMS

We further analyze the number of unmatched responses across different MLLMs for each of the three evaluation criteria: Textual Faithfulness, Frame Consistency, and Video Fidelity.

As shown in Table 6, we observe that both Video-LLaMA and Kangaroo struggle to produce responses in the required format, resulting in null outputs. In contrast, other open-source MLLMs demonstrate superior instruction-following abilities, as indicated by their minimal unmatched responses. However, some closed-source models, such as GPT-4o and Gemini-pro, often fail to provide adequate responses, frequently refusing to process video inputs or extracted frames. These models typically respond with statements like *"I'm unable to process real-world information"* or *"I cannot analyze videos but can assist with text-based tasks."*.

## C MORE DETAILS OF HUMAN ANNOTATION SETUPS

### C.1 EVALUATION GUIDELINES

As shown in Tables 7, 8, and 9, we provide detailed guidelines for evaluators to ensure consistent and accurate scoring. The evaluation is based on three key criteria: Textual Faithfulness, Frame Consistency, and Video Fidelity. For each criterion, we define specific score levels ranging from 1 to 5, with 5 representing the highest quality and 1 representing the lowest. These guidelines are designed to help evaluators assess the quality of the edited videos in a reliable manner.

Table 6: Number of unmatched answers for different evaluators on three criteria.

| Evaluator | Parameters | Textual Faithfulness | Frame Consistency | Video Fidelity |
|---|---|---|---|---|
| *Open-source MLLMs* | | | | |
| TimeChat | 7B | 0 | 0 | 0 |
| Video-LLaMA | 7B | 1280 | 1280 | 1280 |
| Video-LLaMA-2 | 7B | 11 | 209 | 339 |
| Kangaroo | 7B | 1280 | 1280 | 1280 |
| Qwen-VL-Chat | 7B | 0 | 0 | 0 |
| LLaVA-NeXT-Video | 7B | 0 | 0 | 0 |
| LLaVA-NeXT-Video | 32B | 0 | 0 | 0 |
| LLaVA-OneVision | 7B | 0 | 0 | 0 |
| LLaVA-OneVision | 70B | 0 | 0 | 0 |
| VILA | 34B | 5 | 4 | 2 |
| *Closed-source MLLMs* | | | | |
| GPT-4o-0806 | - | 148 | 171 | 104 |
| GPT-4o | - | 3 | 1 | 2 |
| Gemini-pro | - | 187 | 360 | 181 |

Table 7: Guidelines for evaluating Textual Faithfulness on our VEBench.

| Score | Textual Faithfulness |
|---|---|
| 5 | The edited video fully aligns with the text prompt, capturing all details accurately. |
| 4 | The edited video aligns with the text prompt in most aspects, with only a few details not reflected. |
| 3 | The edited video generally aligns with the text prompt, but many details are missing. |
| 2 | The edited video mostly misaligns with the text prompt. |
| 1 | The edited video completely misaligns with the text prompt. |

Table 8: Guidelines for evaluating Frame Consistency on our VEBench.

| Score | Frame Consistency |
|---|---|
| 5 | The frames flow smoothly and continuously without any noticeable jumps. |
| 4 | The continuity between frames is good, with only minimal jumps in a very few scenes. |
| 3 | The continuity between frames is average, with minor jumps in some scenes. |
| 2 | The continuity between frames is poor, with noticeable jumps. |
| 1 | There is no continuity between frames, resulting in a poor viewing experience. |

Table 9: Guidelines for evaluating Video Fidelity on our VEBench.

| Score | Video Fidelity |
|---|---|
| 5 | The video is fully realistic, with excellent visual quality and no noticeable flaws, providing a perfect viewing experience. |
| 4 | The video is close to realistic, with good overall quality and only minor imperfections in rare instances. |
| 3 | The video has slight color distortion and is generally acceptable, but some unnatural elements are still noticeable. |
| 2 | The video has significant color distortion and overall visual quality issues, with noticeable inconsistencies. |
| 1 | The video suffers from severe color distortion, poor visual quality, and weak overall presentation, leading to a very poor viewing experience. |

**Source prompt**

A man carrying a backpack is walking on the mountain, stepping on rocks, with the mountain as the background

**Text prompt**

A woman carrying a backpack is walking on the mountain, stepping on rocks, with the mountain as the background

**Editing target**

Man –> Woman

Figure 7: Screenshot of the human annotation interface, showing the current instance where "Man" in the original video is replaced by "Woman" in the edited video.

## C.2 EVALUATION INTERFACE

As shown in Figure 7 and Figure 8, we develop a specialized tool designed for scoring text-guided video editing tasks. The interface primarily displays the source prompt, text prompt, editing target, and original and edited video, along with basic controls, three criteria scoring buttons, and an expandable detailed data annotation guideline.

Figure 8: Screenshot of the human annotation interface, showing the current instance where "Monkey" in the original video is replaced by "Gorilla" in the edited video.

