# OpenReview forum: "VEBench: Towards Comprehensive and Automatic Evaluation for Text-guided Video Editing"
_ICLR.cc/2025/Conference — Submitted to ICLR 2025_

### Official Review · Reviewer_xiyW · 2024-10-20

**Soundness:** 3
**Presentation:** 2
**Contribution:** 2
**Rating:** 5
**Confidence:** 5

**Summary:**

This paper studies an important task of benchmarking video editing. It collects video clips from the DAVIS dataset and creates text prompts efficiently based on GPT-4V automatically. The data taxonomy / editing types mainly focus on content and style. Besides the benchmark, the authors also developed an automatic metric called VEScore, which aligns better with human ratings and serves as a more robust and scalable automatic method compared to traditional metrics.

A more comprehensive video editing benchmark could contain other popular editing types such as editing motion, emotion, etc. Further, the benchmark could go beyond just text-based video editing, such as providing a reference image and using it to replace the object in the original video. For such reference-image based video editing task, we may need additional metrics for ID preserving of the reference image.

**Strengths:**

1/ This paper studies an important task of benchmarking video editing

2/ Besides the benchmark, the authors also developed an automatic metric called VEScore, which aligns better with human ratings and serves as a more robust and scalable automatic method compared to traditional metrics.

3/ In addition to single object, this benchmark further considers the scenario of multiple objects

**Weaknesses:**

1/ Why only collect videos from DAVIS dataset? Would it be better to include more different sources and domains to increase content diversity?

2/ When reading the introduction, I feel it be more informative to have a teaser image to showcase editing examples along with the textual descriptions in the main text. E.g. When reading "the number of editing targets (Single or Multiple)", a more concrete example like the one in Fig 1 would be helpful.

3/ When proposing multiple targets, understand it contains cases like “brown bear” and “zoo” to be edited as “polar bear” and “Arctic ice”. Are these cases in the benchmarks that edit multiple objects of the same categories? e.g. a video contains multiple "brown bears" and change all or part of them into “polar bear”

4/ In practice, people use not only text for video editing but can incorporate various conditional data. For example, a) provide a reference image of the new object which shall be used to replace the target object in the video. b) besides just change content and style, how about change motion of an object e.g. car moving left in the original video and change it to move right, change walking to other actions. How to make this video editing benchmark more comprehensive and include various editing scenarios, not limited to just text-based editing?

5/ Any specific challenges for editing long video? Would it be interesting to include cases of editing long videos / movie-like videos?

**Questions:**

Kindly see the weakness section

---

> ### Author Response · Authors · 2024-11-20
> **Response to Reviewer xiyW**
>
> 1. Why davis? and include more different sources and domains to increase  diversity
>
> Thank you for your insightful comments on the diversity of our dataset. We chose DAVIS as our primary video source because it is a well-established and widely used testbed in the video editing research community, ensuring that our benchmark aligns with existing standards and facilitates meaningful comparisons with other studies.
>
> However, we understand the importance of incorporating a broader range of video sources to enhance diversity. We are actively working on expanding VEBench to include videos from additional platforms such as YouTube and Videvo, which will allow us to cover a more extensive array of topics and domains, thus improving the dataset’s overall diversity and relevance.
>
> To highlight VEBench's current standing, please refer to the comparison table below, which demonstrates how VEBench surpasses existing datasets in terms of video and text prompt numbers, as well as its unique support for meta-evaluation:
>
> | **Paper**      | **Number of Videos** | **Number of Text Prompts** | **Number of data for Meta-Evaluation** |
> | -------------- | -------------------- | -------------------------- | -------------------------------------- |
> | Dreamix*       | 29                   | 127                        | -                                      |
> | Gen-1*         | unknown              | 35                         | -                                      |
> | Tune-A-Video*  | 42                   | 140                        | -                                      |
> | Text2LIVE*     | 7                    | unknown                    | -                                      |
> | Video-P2P*     | 10                   | unknown                    | -                                      |
> | TGVE 2023*     | 76                   | 304                        | -                                      |
> | VEBench (Ours) | 152                  | 962                        | 1280                                   |
>
> *The asterisk (\*) denotes data reused from source [1], while the dash (\-) indicates that existing benchmarks do not support meta-evaluation, except ours.*
>
> We greatly appreciate your feedback and are committed to making these enhancements. If you have additional suggestions for other potential data sources or domains, we would be happy to consider them to further enrich VEBench. Your feedback is invaluable in helping us improve our work.
>
> *[1]* *Wu, et al. "CVPR 2023 Text Guided Video Editing Competition." \*arXiv preprint\* arXiv:2310.16003, 2023.*
>
> 2. Providing concrete example in introduction
>
> Thank you! We will definitely improve this part.
>
> 3. Edit multiple objects of the same categories
>
> It depends on the edit prompts. For example, "edit the brown bear on the left to polar bear" or "edit all brown bears to polar bears".
>
> Currently, we take each edit target as an exact object in the video instead of objects of the same categories.
>
> We check the dataset and do not meet this problem because the edit prompt is precise and there is no ambiguity in editing an object or objects of the same category.
>
> 4. Incorporate various conditional data for video editing
>
> As mentioned in the Section Introduction, the majority of current Video Editing is Text-guided Video Editing. Therefore, from a practical perspective, we have chosen to evaluate this approach exclusively. We agree other types of video editing are also valuable and will provide a discussion in the related work section.
>
> 5. Specific challenges for editing long video
>
> This is an insightful suggestion. We did not conduct analyses on these cases because current technologies are largely limited to editing short clips, typically within 25 frames. However, we believe this is an important direction and plan to include longer videos from other sources (e.g., YouTube) to better support the evaluation of this case.

---

> > ### Comment · Reviewer_xiyW · 2024-11-26
> >
> > I appreciate authors for the response. Overall I believe this is an important task to work on. But as most concerns remain unresolved and require significant efforts beyond what can be done within this rebuttal period, I would like to encourage the authors to improve the paper via addressing these weaknesses and recycle to the next venue.

---

### Official Review · Reviewer_n2Cu · 2024-10-30

**Soundness:** 3
**Presentation:** 3
**Contribution:** 2
**Rating:** 5
**Confidence:** 5

**Summary:**

This paper presents VEBench, a comprehensive benchmark designed to assess text-guided video editing models, alongside VEScore, an automated scoring system that utilizes Multi-modal Large Language Models (MLLMs) for evaluation. The research evaluates VEBench and VEScore based on criteria such as Textual Faithfulness, Frame Consistency, and Video Fidelity, with the goal of minimizing reliance on expensive human evaluations.

**Strengths:**

1. VEBench provides a substantial and structured benchmark, offering a variety of editing types and criteria that address significant gaps in the evaluation of video editing.
2. The introduction of VEScore showcases the potential for automation in evaluating video edits, presenting preliminary evidence of better alignment with human judgments.
3. The study offers a detailed analysis of current video editing models, highlighting the challenges in maintaining frame consistency and video fidelity, thereby providing valuable insights for future model development.

**Weaknesses:**

1. VEScore relies heavily on prompt-based MLLM outputs without employing fine-tuning or advanced techniques like token probability averaging. This approach is limited by the performance constraints and potential hallucinations of MLLMs, especially in complex scenarios.
2. The study does not thoroughly explore the consistency and reproducibility of prompt engineering, with limited testing across different prompt variations to ensure robustness.
3. The dependence on large-scale models (32B) or expensive closed-source APIs is impractical for most applications, raising concerns about scalability and general accessibility.
4. While VEBench enhances existing benchmarks, it may not significantly advance evaluation methods, as it neither fine-tunes MLLMs on evaluation-specific data nor addresses ongoing issues in evaluating nuanced or multi-target edits.

**Questions:**

1. Given that other recent benchmarks have successfully implemented task-specific training or fine-tuning for video evaluation, could the authors clarify their rationale for opting against these techniques in VEScore?
2. How does the paper address potential variability and hallucination in VEScore’s output, given its reliance on prompt-only evaluation without task-specific fine-tuning?

While VEBench serves as a benchmark that expands the dataset scale and diversity of textual prompts, it does not fundamentally address the critical pain points in video editing evaluation, particularly in objectively assessing detailed changes and stylistic variations in edited videos. Although VEScore demonstrates relatively high correlation with human ratings on certain criteria, its correlation coefficient of approximately 0.49 remains insufficiently significant.

---

> ### Author Response · Authors · 2024-11-19
> **Response to Reviewer n2Cu - Part-1**
>
> Thank you! It seems there is a misunderstanding in the contribution of our work. We kindly note that VEBench is not only the largest-ever video editing benchmark, but it also serves as the first meta-evaluation benchmark in this field to help develop automatic evaluation tools. By carefully designing the evaluation criteria and providing high-quality human annotations on 1280 edited videos, we believe VEBench paves the way and can accelerate the development of more reliable evaluation tools.
>
> Our responses to your questions are as follows.
>
> 1. Without employing fine-tuning or advanced techniques like token probability averaging
>
> We are the first to explore utilizing MLLMs for evaluating video editing. We follow the popular approach [1] to implement our VEScore. We agree that finetuning or other techniques may further enhance the performance. However, finetuning needs to craft corresponding training data, which is non-trivial to obtain and has not been explored in this field. Token probability averaging is also been demonstrated not reliable in expressing uncertainty [2].
>
> *[1] Zheng, et al. "Judging LLM-as-a-Judge with MT-Bench and Chatbot Arena." \*arXiv e-prints\*, arXiv:2306.05685, June 2023.*
>
> *[2] Wang, et al. "Self-Consistency Boosts Calibration for Math Reasoning." \*arXiv preprint\*, arXiv:2403.09849, 2024.*
>
> 2. Consistency and reproducibility of prompt engineering.
>
> Thank you! We will explore paraphrasing the instruction to test the robustness of VEScore.
>
> We did not specially conduct prompt engineering because this necessitates a development set. While we believe this can be potential with the help of VEBench.
>
> Utilizing GPT, we generated multiple paraphrased versions of the instruction, assessed their performance, and analyzed these variations to verify robustness. Due to computational time considerations, this part of our experiments was based on LLaVA-OneVision-7B. In the future, we will evaluate all MLLMs. The outcomes are summarized in the following table:
>
> | Evaluator (LLaVA-OneVision-7B) | TF $r$ | TF $\rho$ | TF $\tau$ | FC $r$ | FC $\rho$ | FC $\tau$ | VF $r$ | VF $\rho$ | VF $\tau$ |
> | ------------------------------ | ------ | --------- | --------- | ------ | --------- | --------- | ------ | --------- | --------- |
> | Original                       | 0.49   | 0.48      | 0.39      | 0.17   | 0.18      | 0.14      | 0.07   | 0.07      | 0.06      |
> | Rewritten Version 1            | 0.46   | 0.44      | 0.37      | 0.16   | 0.14      | 0.12      | 0.11   | 0.11      | 0.09      |
> | Rewritten Version 2            | 0.49   | 0.48      | 0.40      | 0.23   | 0.24      | 0.19      | 0.12   | 0.15      | 0.12      |
> | Rewritten Version 3            | 0.50   | 0.49      | 0.40      | 0.25   | 0.26      | 0.21      | 0.14   | 0.16      | 0.13      |
> | Rewritten Version 4            | 0.50   | 0.48      | 0.40      | 0.18   | 0.17      | 0.14      | 0.10   | 0.11      | 0.09      |
> | Rewritten Version 5            | 0.48   | 0.46      | 0.38      | 0.24   | 0.25      | 0.20      | 0.12   | 0.16      | 0.13      |
>
> *TF, FC, and VF denote Textual Faithfulness, Frame Consistency, and Video Fidelity, respectively. And the metrics $r$(↑), $\rho$(↑), and $\tau$ (↑) correspond to Pearson Correlation, Spearman Correlation, and Kendall’s Tau, respectively.*
>
> We have included a calculation of the mean and standard deviation for each evaluation metric as follows:
>
> - TF $r$: 0.49 ± 0.02
>
> - TF $\rho$: 0.47 ± 0.02
>
> - TF $\tau$: 0.39 ± 0.01
>
> - FC $r$: 0.20 ± 0.04
>
> - FC $\rho$: 0.21 ± 0.05
>
> - FC $\tau$: 0.17 ± 0.04
>
> - VF $r$: 0.11 ± 0.02
>
> - VF $\rho$: 0.13 ± 0.04
>
> - VF $\tau$: 0.10 ± 0.03
>
> These statistics were derived from multiple repeated experiments. We calculated the mean and standard deviation for each metric to assess consistency and robustness under various rewritten prompt versions. We found that overall, MLLMs are relatively robust to changes in prompts, with relatively small standard deviations and minor differences between the means and the values reported in our paper. Additionally, we have provided the specific prompts used in the article in the Supplementary Material and plan to continue open-sourcing these prompts to ensure the reproducibility of all experimental results.
>
> 3. The dependence on large-scale models (32B) or expensive closed-source APIs is impractical
>
> We agree that the VEScore is more expensive than traditional metrics based on CLIP. However, it is still much cheaper than human evaluation.
>
> Besides, we have demonstrated that some small models (e.g., LLaVA-OneVision-7B) also perform well on some criteria. We believe evaluation can be more efficient with the rapid development of small-scale open-source LLMs.

---

> > ### Author Response · Authors · 2024-11-19
> > **Response to Reviewer n2Cu - Part-2**
> >
> > 4. VEBench may not significantly advance evaluation methods
> >
> > As discussed above, VEBench also serves as a meta-evaluation benchmark, which is designed to develop automatic evaluation metrics. VEScore has also not been explored before in this field. We have demonstrated that it significantly outperforms traditional methods. We believe both can accelerate the development of more reliable evaluations.
> >
> > 5. Implement task-specific training or fine-tuning for video evaluation
> >
> > Using finetuning for evaluation is a promising approach. However, evaluating an edited video has to consider the original caption, editing prompt, and edited video altogether. Crafting corresponding training data is non-trivial and has not been explored. Our work with VEBench is primarily a Meta-Evaluation Benchmark designed to assess various MLLMs in Video Editing evaluations, including those specifically fine-tuned for this task. Additionally, as of now, it seems that there hasn't been much exploration of Video Editing Evaluation based on task-specific fine-tuning.
> >
> > 6. Address potential variability and hallucination in VEScore’s output
> >
> > We observe some unfaithfulness in VEScore results as introduced in Sec B3. This can be a critical issue influencing the performance. Nevertheless, VEScore still obviously outperforms traditional metrics. We leave further improvement in future.

---

> > > ### Comment · Reviewer_n2Cu · 2024-11-26
> > >
> > > Thank you for addressing the comments. Given the scope of remaining issues, I suggest strengthening the paper further and submitting to a future venue where you'll have more time to incorporate these important improvements.

---

### Official Review · Reviewer_ynqK · 2024-11-02

**Soundness:** 2
**Presentation:** 3
**Contribution:** 2
**Rating:** 3
**Confidence:** 4

**Summary:**

The paper presents VEBench, a comprehensive meta-evaluation benchmark designed for assessing automatic evaluation metrics in text-guided video editing tasks.  VEBench includes 152 video clips and 962 text prompts categorized by editing type and target complexity. Additionally, the benchmark involves three main evaluation criteria—Textual Faithfulness, Frame Consistency, and Video Fidelity—and employs VEScore. It is a novel benchmark, but some prons should be solved in the rebuttal process.

**Strengths:**

1. Authors meticulously curated a dataset, verified its quality, and categorized it based on editing complexity, enhancing its robustness.
2. The paper thoroughly describes the data collection, taxonomy, annotation processes, and experimental setup.

**Weaknesses:**

1. The amount and diversity of data is a major concern for me. Selecting only 152 videos from the DAVIS2017 and 2019 datasets, and each video has only 25 frames, seems to me to be insufficient. Although the author has tried his best to further classify and study the data in a fine-grained manner, the limited size of the data still makes this part of the data not representative enough.

2. The author did not provide enough examples throughout the article to demonstrate the dataset. With only some text descriptions and some statistical analysis, it is still difficult for me to intuitively see the challenge and diversity of the data itself. Therefore, the appendix needs more presentations on video sequences and a more fine-grained introduction to videos.

3. I would also like more examples, including good and bad cases. I think evaluation is crucial for the future development of algorithms, and bad cases are the key to further exploring the bottlenecks of algorithms. As a Benchmark article, the author not only needs to introduce the evaluation work itself clearly, but also needs to clearly give a very fine-grained explanation and analysis to explain what advantages and disadvantages the currently tested methods have in specific situations, and to illustrate with specific picture sequences. Unfortunately, the current version does not have these contents, so it is more like a technical report than an academic paper.

4. Finally, I would also like to know the more specific processes and standards used by the authors when selecting human subjects, such as the basic situation of these subjects, whether their cognitive levels are consistent, and whether their ability to understand the tasks is similar, to ensure that the evaluation results provided by the human subjects are valid.

**Questions:**

Please see the weaknesses.

---

> ### Author Response · Authors · 2024-11-20
> **Response to Reviewer ynqK**
>
> Thank you! It seems there is a misunderstanding in the contribution of our work. We kindly note that VEBench is the largest-ever video editing benchmark. More importantly, it serves as the first meta-evaluation benchmark in this field to help develop automatic evaluation tools. This is an emergent requirement due to the heavy reliance on human evaluation to date. We are also the first to demonstrate the potential of utilizing MLLMs as evaluators for video editing. We believe our work can benefit the community. There are the responses to your questions. Hope these can address your concerns.
>
> 1. The amount and diversity
>
> Thank you for highlighting this aspect. We acknowledge that while VEBench is not as large as some general-purpose datasets, it is significantly larger than other specialized video editing datasets, as outlined in the related works section. More importantly, VEBench serves as the first meta-evaluation benchmark for video editing, which we believe will be instrumental in the development of reliable automatic evaluation tools, thus alleviating the current reliance on extensive human evaluation efforts.
>
> | **Paper**      | **Number of Videos** | **Number of Text Prompts** | **Number of data for Meta-Evaluation** |
> | -------------- | -------------------- | -------------------------- | -------------------------------------- |
> | Dreamix*       | 29                   | 127                        | -                                      |
> | Gen-1*         | unknown              | 35                         | -                                      |
> | Tune-A-Video*  | 42                   | 140                        | -                                      |
> | Text2LIVE*     | 7                    | unknown                    | -                                      |
> | Video-P2P*     | 10                   | unknown                    | -                                      |
> | TGVE 2023*     | 76                   | 304                        | -                                      |
> | VEBench (Ours) | 152                  | 962                        | 1280                                   |
>
> *The asterisk (\*) denotes data reused from source [1], while the dash (\-) indicates that existing benchmarks do not support meta-evaluation, except ours.*
>
> As illustrated in the comparison table, VEBench surpasses existing datasets in both video and text prompt quantities. Additionally, we have selected the DAVIS dataset due to its prominence in previous studies, ensuring our work aligns with established research. Nevertheless, we are actively working on incorporating videos from other sources like YouTube and Videvo to further enhance the diversity of VEBench. We appreciate your suggestions and are open to any additional feedback you might have on how we can further improve this aspect of our work.
>
> *[1]* *Wu, et al. "CVPR 2023 Text Guided Video Editing Competition." \*arXiv preprint\* arXiv:2310.16003, 2023.*
>
> 2. Each video has only 25 frames
>
> Sorry for the misunderstanding. VEBench contains video with frames ranging from 25 to 127. Due to the constraints of popular video editing models, we only adopt 25 frames as the model inputs for evaluation. We will improve the corresponding part in the manuscript.
>
> 3. Lacking examples of VEBench and VEScore
>
> Thank you! We will provide more typical cases of our dataset and evaluation results in the revision. We kindly note that statistical results (Table 6, Figure 6) can demonstrate the diversity of VEBench. Besides, an error analysis about VEScore using different MLLMs is also provided in Sec B3.
>
> 4. Specific processes and standards when selecting human subjects
>
> Thank you for your feedback. We recruited annotators who hold a master's degree and have a background in multimodal research. Before the official annotation process began, we conducted a one-week trial phase to evaluate accuracy and consistency. This was followed by a two-week formal annotation period using our annotation platform. We ensured that none of the annotators had any conflicts of interest, and we routinely cross-checked their annotations for consistency to maintain high-quality standards.

---

> > ### Comment · Reviewer_ynqK · 2024-11-26
> >
> > Dear Authors,
> >
> > I have carefully read all the comments from other reviewers. Thanks a lot for your detailed responses.
> >
> > While I appreciate the motivation behind your work and agree that benchmarking video editing evaluation is a critical and underexplored area, I believe substantial gaps in the current submission need to be addressed to strengthen its contributions.
> >
> > Firstly, the proposed benchmark is indeed important and relevant. However, as an evaluation environment for challenging tasks like text-guided video editing, more effort is needed to articulate and address this task's unique characteristics and difficulties. It is crucial to specify how this task differs from others and why these differences make it particularly challenging. Small nuances in task definitions can lead to significant changes in evaluation strategies, and these nuances must be explicitly discussed.
> >
> > Secondly, the construction of the dataset and evaluation environment requires more careful consideration. Merely using existing datasets, even if re-organized from a new perspective, might not suffice unless the scale or diversity is significantly expanded. The current dataset primarily relies on a limited number of existing sources, which constrains its novelty and comprehensiveness. A stronger effort to demonstrate unique insights in dataset construction is necessary.
> >
> > Thirdly, the evaluation itself needs more granularity and diversity:
> >
> > - Diversity of Tested Models: Consider incorporating a wider variety of models, including those that may not strictly align with the current task but provide alternative approaches.
> > - Comparative Experiments: Design experiments that compare different evaluation methods under varying conditions to highlight the robustness and generalizability of your proposed metrics.
> > - Evaluation Metrics: Provide a deeper discussion of how your chosen metrics address the task's unique challenges and explore the potential trade-offs or biases inherent in these metrics.
> > - Results Analysis: Both qualitative and quantitative results need more thorough analysis. Use case studies, detailed failure analysis, and interpretive visualizations to substantiate your claims about task difficulty and evaluation effectiveness.
> >
> > Finally, while the topic and motivation are promising, the overall scope of the work feels limited for a benchmark paper. Building a comprehensive benchmark involves demonstrating novelty in environment construction, metrics design, and evaluation robustness. These aspects need significant enhancement to align with the importance of the proposed task.
> >
> > I encourage you to continue working on this project and further refine it. Strengthening the dataset, metrics, and analysis will make this work a more robust and impactful contribution to the field.
> >
> > Thus, I will keep my score now.

---

### Official Review · Reviewer_ErM7 · 2024-11-03

**Soundness:** 3
**Presentation:** 3
**Contribution:** 2
**Rating:** 5
**Confidence:** 2

**Summary:**

This paper presents a comprehensive meta-evaluation benchmark for text-guided video editing, and proposes VEScore for automatic evaluation. It addresses the lack of reliable metrics in video editing evaluation. Experiments show challenges for current models and the potential of VEScore.

**Strengths:**

1. The proposed benchmark VEBench is the largest-ever in this field, providing a rich set of data including diverse video clips, text prompts, and human annotations, which enables in-depth analysis of video editing models.

2. The proposal of using MLLMs as evaluators (VEScore) is innovative and shows great potential. It outperforms traditional metrics in correlating with human judgments, especially in evaluating Textual Faithfulness.

3. The experiments reveal valuable insights about the performance of current video editing models, such as their struggles with certain editing types.

**Weaknesses:**

1. Although VEBench contains 152 video clips and 962 text prompts, compared to some large-scale general-purpose datasets, its data diversity may still be insufficient. For example, the data sources mainly focus on the DAVIS dataset, which may have limitations in terms of video scenes, content themes, shooting styles, etc., and may not comprehensively cover all types of video editing needs in the real world.

2. Although the paper mentions using MLLMs for evaluation, it lacks in-depth analysis of the specific strengths and limitations of different MLLMs when handling video editing evaluation tasks.

**Questions:**

Please see Weaknesses

---

> ### Author Response · Authors · 2024-11-19
> **Response to Reviewer ErM7**
>
> We would like to thank the reviewer for their valuable feedback and positive comments. We appreciate the recognition of the innovative aspects and potential contributions of our work. Below, we address the concerns raised:
>
> 1. The amount and diversity of VEBench.
>
> Though VEBench is smaller than the general-purpose dataset, it is still much larger than other datasets in this field as described in the related work section.
>
> More importantly, please note that VEBench also serves as the first meta-evaluation benchmark for video editing. We believe VEBench can help develop more reliable automatic evaluation tools to address current heavy human efforts.
>
> A comparison with the existing dataset is shown in the table below:
>
> | **Paper**      | **Number of Videos** | **Number of Text Prompts** | **Number of data for Meta-Evaluation** |
> | -------------- | -------------------- | -------------------------- | -------------------------------------- |
> | Dreamix*       | 29                   | 127                        | -                                      |
> | Gen-1*         | unknown              | 35                         | -                                      |
> | Tune-A-Video*  | 42                   | 140                        | -                                      |
> | Text2LIVE*     | 7                    | unknown                    | -                                      |
> | Video-P2P*     | 10                   | unknown                    | -                                      |
> | TGVE 2023*     | 76                   | 304                        | -                                      |
> | VEBench (Ours) | 152                  | 962                        | 1280                                   |
>
> *The asterisk (\*) denotes data reused from source [1], while the dash (\-) indicates that existing benchmarks do not support meta-evaluation, except ours.*
>
> We chose DAVIS as a video source because it is the most popular testbed in previous studies. By statistics, VEBench covers a wide range of topics as shown in Table 5 and Figure 6. This demonstrates the diversity of VEBench.
>
> Some works also adopt videos from YouTube and Videvo, but their scales are often limited. We are working on appending these videos of different sources to VEBench.
>
> We will improve the corresponding part in the manuscript to avoid misunderstanding and appreciate it if you could provide any further suggestions.
>
> *[1] Wu, et al. "CVPR 2023 Text Guided Video Editing Competition." \*arXiv preprint\* arXiv:2310.16003, 2023.*
>
> 2. In-depth analyses of using MLLMs as evaluators
>
> **Table 4** presents the performance of MLLMs across various dimensions. Besides, In **Table 6**, we provide an error analysis about different MLLMs (This part is put in the Appendix due to page limit). We notice that one important issue is the capability of faithfully following our instructions due to the capabilities of weak MLLMs and safety concerns of close-sourced models. Nevertheless, VEScore still performs significantly better than traditional metrics. We believe this approach has potential and can be further improved with the guidance of meta-evaluation annotations in VEBench.

---

> ### Comment · Reviewer_ErM7 · 2024-11-27
>
> Thank the author for the rebuttals. This work has some existing issues that require further attention, but with some improvements, I believe it has the potential to reach the necessary standard for acceptance.

---

### Meta-Review · Area_Chair_KNzd · 2024-12-18

**Metareview:**

This paper proposed a new benchmark for text-guided video editing named VEBench. It collects 152 video clips with 962 carefully designed prompts. An LLM-based auto evaluation VEScore is proposed. The best performed video editing models considered in this paper achieved 3.18 out of 5 on VEScore, which shows the challenges of the proposed benchmark. Reviewers agree that this paper targets an important question and has great potential, however, the insufficiency and lack of diversity of the benchmark is the major concern raised by the reviewers.

Strengths:
1. Reviewers agree that this paper shows great potential to apply MLLMs as evaluators for video editing.
2. The proposed benchmark is the largest-ever in this field.
3. This paper provides some insights on the challenges of existing video editing models.

Weakness:
1. The biggest weakness which agreed by all reviewers is the insufficiency and lack of diversity for the videos collected. Though the benchmark is already the largest of its kind, only 152 videos all from DAVIS is still not sufficient to be a solid benchmark.

**Additional Comments On Reviewer Discussion:**

The major concern from most reviewers (ErM7, ynqK and xiyW) are the insufficiency (only 152 videos) and lack of diversity (all videos are from DAVIS dataset) of the proposed benchmark. The authors mentioned that the proposed one is already the largest of its kind by comparing with some existing benchmarks and argued that DAVIS is well-established and widely used testbed. However, this cannot fully address the concern from the reviewers.

After discussion, reviewers agree that this paper has great potential but their concerns are not fully addressed and recommend the author to further improve the paper.

---

### Decision · Program_Chairs · 2025-01-22

Reject